# A Multimodal Label Forecasting Method for Aperiodic Visuo-Motor Time Series

## Abstract

Deep learning models have been increasingly applied to Time Series Forecasting (TSF) in recent years. Transformer-based and MLP-based models have both been used effectively on many real-world TSF regression benchmarks, and there is ongoing debate as to which family of methods is best. While these benchmarks have drawn much attention, it is also worth noting that many current datasets and methods assume approximate periodicity in the time series. In this work, we focus on a new TSF task without periodicity: anticipating falls during humanoid locomotion, on the basis of egocentric vision and proprioception. When the locomotion trajectories are sufficiently diverse, periodicity is violated. We contribute two new benchmark datasets (one from simulation, one from real hardware), showing that periodicity is violated and recent deep TSF methods struggle on these benchmarks. We also propose a novel deep learning architecture that exploits both endogenous and exogenous variables and a training process that rigorously enforces i.i.d sampling of training examples. Our results show statistically significant improvement over prior art in multiple experimental conditions, by 12.73% or more on the real data and 10.40% or more on the simulation data. Code and datasets will be available upon acceptance.

## 1 Introduction

The long-standing field of time-series forecasting (TSF) has recently witnessed growing interest from the deep learning community. Benchmarking datasets have been published across multiple domains such as temperature and climate (Zhou et al., 2021b; Kolle, 2024), illness (the influenza-like illness dataset (CDC, 2020)), traffic (the PeMS dataset (Song et al., 2020) and (CDT, 2018)), electricity (Trindade, 2015) and finance (Lai et al., 2018). Various deep learning models, from recurrent neural networks (RNNs) (Jordan, 1997) to Transformer-based models (Vaswani et al., 2017), have been introduced to learn the nonlinear relationships in TSF datasets, with techniques such as time stamp encoding and periodicity segmentation.

While such works have achieved impressive results, they are subject to certain limitations, which lead us to propose a new challenge in this area. First, nearly all existing deep TSF benchmarks have clear periodicity, as illustrated in Figure 1 (and shown more comprehensively in Appendix Figure 5). One notable exception is the non-stationary Exchange Rate dataset, but it is also less frequently used by recent works. Second, existing tasks in TSF fall into two categories: regression and classification. Regression predicts future values for the time-series variables, whereas standard classification predicts a single label for an entire time-series. However, certain real-world applications would benefit from the prediction of future, per-time-step class labels. Even though existing models exhibit satisfying performance in standard TSF classification, we are interested in their effectiveness at *forecasting* class labels at time-points that have not yet been observed. Furthermore, recent studies (Wang et al., 2024b; Das et al., 2023) show that TSF performance can be enhanced by incorporating exogenous variables and multiple input modalities. However, existing methods are mostly unimodal and do not leverage exogenous variables (aside from time-stamp encodings for attention-based models).

Based on the foregoing, we propose a new benchmark challenge from the robotics domain, and establish a new baseline model to solve it, named EMP because it leverages **E**gocentric vision, **M**otion planning and **P**roprioception. Our contributions are summarized as follows:

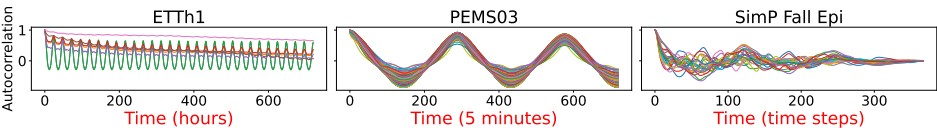

Figure 1: Autocorrelation plots for two standard TSF benchmark datasets as well as our SimP dataset, described in the text. Autocorrelations near 1 indicate periodic signals. Each time step is roughly 33 ms for our SimP dataset. See Figure 5 for similar examples from other benchmarks.

1. We propose a new fine-grained TSF classification task, namely, predicting future per-time-step class labels on aperiodic datasets with exogenous variables.

2. We collect and publish one real and one simulated dataset for this task containing joint angles and head camera video streams recorded from a Poppy® Humanoid (Lapeyre et al., 2013). Each frame has its own per-time-step label indicating whether the robot has fallen. We also confirm that our datasets exhibit less periodicity relative to existing benchmarks (Figures 1 and 5). The exogenous variables are the planned (not actual) joint angle trajectories, whose future values are known.

3. We present the EMP baseline model for this new benchmark, which consistently outperforms existing state-of-the-art (SOTA) models developed for existing benchmarks.

## 2 RELATED WORK

### 2.1 TRADITIONAL AND RNN TSF MODELS

Traditional models, e.g., ARIMA, are less competitive than emerging deep models on non-linear TSF datasets. Early RNNs like Echo State Networks (ESNs) (Jaeger & Haas, 2004) and Long Short-Term Memories (LSTMs) (Hochreiter & Schmidhuber, 1997) have proven to be strong baselines on such datasets. Recently, LSTNet (Lai et al., 2018) combined classical Autoregressive (AR) modeling with deep learning, and Temporal Hierarchical One-Class (THOC) Network (Shen et al., 2020) presented a dilated RNN architecture with skip connections for temporal anomaly detection tasks. Techniques used in early TSF models, such as time series decomposition and moving averages (MAs), have also proven effective in deep learning approaches (Wu et al., 2021; Wang et al., 2022). Dey et al. (2022) adopts a semi-supervised pipeline based on the random forest algorithm (Breiman, 2001) to tackle slip prediction tasks on a quadruped robot.

### 2.2 TRANSFORMER-BASED MODELS AND ALTERNATIVES

Several recent SOTA TSF approaches are transformer-based (Vaswani et al., 2017), such as Informer (Zhou et al., 2021a), Autoformer (Wu et al., 2021), and PatchTST (Nie et al., 2022). Similar to the standard language modeling approach, temporal order information is provided to these models with trainable time-stamp encodings. Transformers have also been applied to frequency-domain representations; for example FEDFormer (Zhou et al., 2022) introduces a discrete Fourier transform into its self-attention mechanism. TimesNet (Wu et al., 2022a), TimeMixer (Wang et al., 2024a) and TimeDART (Wang et al., 2025) incorporate a multiperiodicity analysis to improve performance. TimeXer (Wang et al., 2024b) models time stamps as exogenous variables and outperforms various baselines, although use of more physically meaningful exogenous variables is underexplored. Whereas most of the aforementioned models focus on regression, TimesNet and FlowFormer (Wu et al., 2022b) conduct extensive experiments on TSF classification tasks.

Despite their SOTA performance, transformer-based models are very computationally expensive and several researchers have questioned whether such complexity is necessary. DLinear (Zeng et al., 2023) is a simple yet competitive model on several benchmarks. STD-MAE (Gao et al., 2023), a self-supervised masked autoencoder (MAE), reduces computation by decoupling reconstruction along spatial and temporal dimensions. Another simple yet efficient MLP model, TiDE (Das et al., 2023), augments a linear encoder with time-derived features such as minutes of the hour.

## 2.3 MULTIMODAL TSF

Multimodal input has been a recent focus in foundation models, including for TSF tasks. Time-LLM (Jin et al., 2024) uses pre-trained Llama-7B (Touvron et al., 2023) as the default backbone and reprograms patch embeddings to align two modalities. VisionTS (Chen et al., 2024a) combines vision and other sensor data to train a universal forecasting foundation model with strong zero-shot performance. Time-VLM (Zhong et al., 2025) employs pre-trained vision-language models to extract visual and textual features from time series.

## 2.4 EGOVISION-AUGMENTED FALL PREDICTION

In a separate line of research, several models have been designed specifically for visuo-motor time-series tasks. EgoEgo (Li et al., 2023) tackles the regression task of predicting full-body pose from egocentric visual input by first predicting head camera pose and employs the DROID-SLAM technique (Teed & Deng, 2021); Marepo (Chen et al., 2024b) and transformer-based models (Shavit et al., 2021) are other approaches for this task. Although not the focus of this paper, there is also a large body of work using egovision not for forecasting, but for control – e.g., Agarwal et al. (2023).

## 3 METHODOLOGY

This section details our proposed TSF benchmark challenge, namely per-time-step class label forecasting with visuo-motor data, and our proposed EMP model for this task, illustrated in Figure 2.

### 3.1 PROBLEM STATEMENT

Let $\boldsymbol{X}_{t-N:t} \in \mathbb{R}^{N \times C_{\text{endo}}}$ denote a time-series with $C_{\text{endo}}$ endogenous variates per time-step, observed for the past $N$ time-steps up to time $t$. The per-time-step class forecasting task is to predict a future categorical label $L_{t+P} \in \mathbb{Z}$, occurring $P$ time-steps in the future, where $P \in \mathbb{N}$. A time-series $\boldsymbol{Y}_{t-N:t+P} \in \mathbb{R}^{(N+P) \times C_{\text{exo}}}$ of $C_{\text{exo}}$ exogenous variates, for the past $N$ and future $P$ time-steps, can also be provided. For example, in a robot control task, the exogenous time-series can be a planned joint trajectory, which is constructed by a motion planner and hence known to the robot before it is fully executed. The time series can include multiple modalities, such as egovision and joint angles, in which case $C_{\text{endo}}$ denotes the total number of variates across all modalities (e.g. number of joints plus the size of the image raster).

### 3.2 EMP ARCHITECTURE

Our proposed architecture, EMP, tackles the problem stated above for the case of visuo-motor time-series data in a robotics context. The architecture, illustrated in Figure 2, combines three data streams from two modalities: Egocentric visual input (collected from the robot's head camera), and joint trajectory information (both actual and planned, collected from the robot's joint sensors and motion planner, respectively). Three encoders are used to extract features from their respective input streams, and then the flattened, concatenated features are passed through one more linear layer for the final class prediction. Details on each encoder are provided in the following subsections. Hyperparameters are given in the Appendix.

#### 3.2.1 MOTION ENCODER

The motion encoder takes actual joint positions as input, and outputs learned spatio-temporal representations. The input stream is a batch of 2D matrices, each of shape $\mathbb{R}^{N \times C}$, where $N$ is the number of time-steps and $C$ is the number of joints. We let $\mathbf{M}$ denote this batched input where $M_{b,t,j}$ is the $j$th joint angle at the $t$th time-step in the $b$th input sequence of the batch.

Inspired by Guo et al. (2023), we use a similar but more lightweight model which separates temporal and spatial integration layers. Formally, letting $\boldsymbol{U}$ and $\boldsymbol{W}$ denote appropriately shaped matrix parameters (either $N \times N$ or $C \times C$), note that left multiplication $\boldsymbol{U} \cdot \mathbf{M}_{b,:,:}$ integrates over time, applying the same transformation to each column (i.e., the time-series for each joint), whereas right multiplication $\mathbf{M}_{b,:,:} \cdot \boldsymbol{W}$ integrates over "space" applying the same transformation to each row (i.e.,

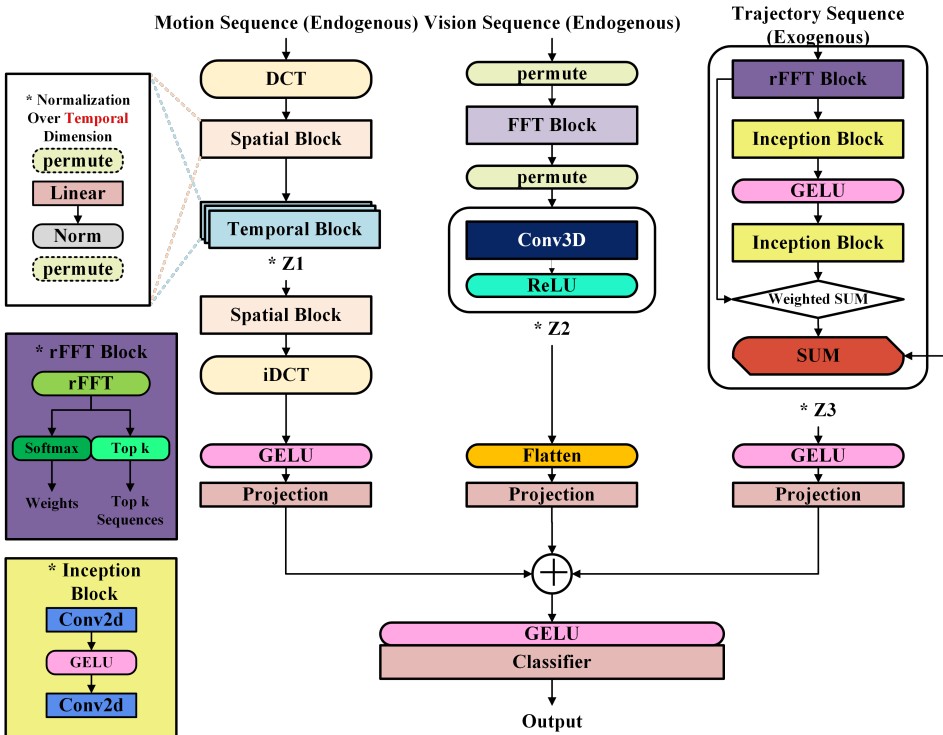

Figure 2: The overview of our approach, which incorporates proprioception and egocentric vision, endogenous and exogenous variables for TSF. Spatial and temporal blocks have similar structure, where permutation is applied for rearranging dimensions. The rFFT block and inception blocks are detailed at the bottom left corner. Blocks in the same color refer to the exact same layers in the PyTorch framework (Paszke et al., 2017).

the full joint configuration for each time-step). Having each model layer integrate over one dimension or the other (but not both) makes the model less computationally intensive. Both types of integration can be implemented using a linear layer if the input series are transposed accordingly.

The first layer of the motion encoder applies a Discrete Cosine Transform (DCT) (Ahmed et al., 2006), to each joint's individual time-series before feeding to a spatial block. The DCT can be implemented as a batched matrix multiplication of an $N \times N$ DCT matrix $\boldsymbol{D}$ with each $N \times C$ input sequence, i.e. $\boldsymbol{D} \cdot \mathbf{M}_{b,:,:}$, where:

$$\boldsymbol{D}_{i,j} = \frac{1}{\sqrt{1 + \delta_{i,0}}} \sqrt{\frac{2}{N}} \cos\left[\frac{\pi}{N}(j + \frac{1}{2})i\right] \tag{1}$$

with $\delta_{i,j}$ denoting the Kronecker delta

$$\delta_{i,j} = \begin{cases} 1, & \text{if } i = j \\ 0, & \text{otherwise.} \end{cases} \tag{2}$$

Likewise, the final layer uses the inverse DCT matrix $\boldsymbol{D}^{-1}$ to transform the processed information back to the original domain. In between the first and last layer is a symmetric Multi-Layer Perceptron (MLP). We add two spatial blocks right after the DCT transformation and before the iDCT transformation for the spatial dimension. In between the two spatial blocks, a stack of temporal blocks are introduced to operate on the temporal dimension. Unlike the previous work, each block in our encoder contains a linear layer and an instance normalization layer over the spatial dimension, which turns out to be the optimal choice for our dataset. For example, the forward pass of the

temporal block is:

$$\boldsymbol{\mu} = \frac{1}{C} \sum_{i=1}^{C} \hat{\mathbf{M}}_{:,i,:}^{\top} \tag{3}$$

$$\boldsymbol{\sigma} = \frac{1}{C} \sum_{i=1}^{C} (\hat{\mathbf{M}}_{:,i,:}^{\top} - \boldsymbol{\mu})^2 \tag{4}$$

$$\tilde{\mathbf{M}}_{:,i,:}^{\top} = \frac{\hat{\mathbf{M}}_{:,i,:}^{\top} - \boldsymbol{\mu}}{\sqrt{\boldsymbol{\sigma}^2 + \epsilon}}, \tag{5}$$

where $\hat{\mathbf{M}}$ is the input to the temporal block's normalization layer, $\epsilon = 10^{-5}$ avoids zero division, $C$ refers to the number of joints, and $\hat{\mathbf{M}}_{:,i,:}^{\top}$ refers to the $i$th joint in the transposed motion sequence $\hat{\mathbf{M}}^{\top}$ with the last two dimensions interchanged. The complete forward propagation for a motion sequence can be written as:

$$\mathbf{M}^{\text{out}} = \boldsymbol{D}^{-1}\text{Spa}(\text{TepMLP}(\text{Spa}(\boldsymbol{D} \cdot \mathbf{M})), \tag{6}$$

where $Spa$ and $TepMLP$ refer respectively to the spatial block and an MLP consisting of multiple temporal blocks.

### 3.2.2 VISION ENCODER

Egocentric visual input is stored in a batch of RGB image sequences $\mathbf{V}$ with shape $(B, 3, N, H, W)$, where $B$ is batch size, 3 comes from the three color channels, $N$ is again the number of time-points and $H$ and $W$ are image height and width. Our robot looks forward while walking and therefore it cannot see any part of itself, which imitates human beings walk (see Figure 8 in Appendix C). Our vision encoder is based on the intuition that falls are manifested as rapid changes to large portions of the field of view, as opposed to small localized details. Hence, we leverage the low-spatial-frequency domain of the images since it captures large structures while ignoring small details.

The image sequence is transformed to the spatial-frequency domain by a Fast Fourier block where a square mask $\mathbf{K}_a$ is defined by the given low-pass frequency threshold $a$. The threshold is set to $a = 30$ in our implementation which satisfies

$$0 < a < \frac{1}{2}\min(H, W). \tag{7}$$

The low-pass filter block, denoted $F(\cdot)$ is implemented as:

$$F(\mathbf{V}) = \text{ifft}(\text{fft}(\mathbf{V}) \odot \mathbf{K}_a), \tag{8}$$

where fft and ifft refer to Fast Fourier Transform and its inverse, which operate on the last two dimensions of their input, and $\odot$ indicates the Hadamard product.

However, the Fast Fourier block does not change the dimensionality of the input sequence, and the temporal dependencies between consecutive filtered images are not processed by our model yet. We achieve such processing with a block of 3D convolutional layers, with multi-scale kernels and ReLU activations layers, denoted Conv3D:

$$\mathbf{V}^{\text{out}} = \text{Conv3D}(F(\mathbf{V})). \tag{9}$$

### 3.2.3 TRAJECTORY ENCODER

Motivated by the fact most existing TSF datasets do not provide exogenous variables other than time stamps and most deep models (Das et al., 2023; Wang et al., 2024b) have to incorporate time stamps as exogenous variables, our dataset and model involve a bonafide, physically meaningful exogenous variable: the planned joint trajectory $\mathbf{T} \in \mathbb{R}^{B \times (N+P) \times C}$. Apart from the current positions and visual cues that are ever changing during robotic humanoid walking, every episode is given a different trajectory as the exogenous variable. Since the trajectory is planned in advance, this input sequence does not change during the movement. Including trajectories as exogenous variables is useful because the quality of the motion plan influences the likelihood of a fall in the near future.

Even though there is limited apparent periodicity in the observed joint angles (see Figure 5), due to sensor/actuator noise and environmental perturbations, planned trajectories may have more reliable periodicity (e.g., in a repetitive gait) that our model can exploit. Note that the motion plan can be periodic, whereas the actual observed joint positions, visual input, and importantly the fall/stand labels, are not periodic. This is done with a series of convolutional blocks to filter the frequencies with the top $k$ amplitudes using Fast Fourier Transformation. Following the same method as Wang et al. (2024a), the filtered frequencies are reshaped into 2D tensors for 2D convolutional layers to capture interperiod- and intraperiod-variations. The convolutions are applied within a residual layer:

$$\mathbf{T}^{\text{out}} = \mathbf{T} + \text{ConvBlk}(\sum_{i=1}^{k} w_i * \text{Fre}(\mathbf{T})), \tag{10}$$

where $\text{Fre}(\cdot)$ stands for the filtering step for the top $k$ frequencies, ConvBlk refers to the 2D convolutional layers with multi-scale kernels followed by GELU activation layers, and the weights $w$ are obtained by passing the top $k$ amplitudes from the rFFT output through a softmax.

## 4 EXPERIMENTS

We evaluated EMP and several SOTA TSF models on a fine-grained binary classification task: Predicting whether or not a humanoid robot will fall $P$ timesteps into the future. In contrast with existing TSF regression benchmarks where outputs are easy to interpret (temperature, exchange rate, etc.), robot joint angles are a less interpretable and actionable output than a fall prediction, which motivates our proposed per-time-step class label forecasting task. This section describes our benchmark datasets, experimental comparison of EMP and SOTA baseline models, and ablation studies on the different modality encoders in EMP.

### 4.1 DATASETS

We collected two datasets using the Poppy humanoid robot, one with real hardware (RP) and one in simulation (SimP). We use rejection sampling to ensure that class labels are balanced and strictly enforce i.i.d. training samples, meaning that each input sequence in the dataset is drawn from a separate episode, as opposed to using sliding windows that reuse data from the same episodes.

#### 4.1.1 RP DATASET

The RP dataset has approximately 16 fps and contains 110 episodes collected from three locations on our institution's campus: (1) an office, (2) a computer laboratory, and (3) a hallway. Poppy takes 6 foot-steps forward in each episode, using random perturbations of a hand-designed walking gait. We recorded by hand which of the 6 foot-steps, if any, contained a fall, and labeled all frames within that and subsequent steps as falls. The floor is carpeted in the office and laboratory but tiled in the hallway; the latter is more challenging since Poppy falls more often on the slippery floor. Episodes collected in the office and laboratory are used for training and validation, and hallway for testing.

#### 4.1.2 SIMP DATASET

We collect 2K falling and walking episodes with 30 fps in a virtual PyBullet (Coumans & Bai, 2016–2021) environment with position control. Trajectory planning is based on Rapidly-exploring Random Trees (LaValle, 1998) and produces a more diverse trajectory set than in the real hardware experiment. We also add perturbations to trajectory waypoints to induce falls. The perturbation is the product of an amplifier factor following logistic growth and a noise factor $\eta$, sampled from a Gaussian distribution centered at 0 mean with standard deviation, $\sigma$, iterating over $[0.005, 0.010, 0.035, 0.040]$. Formally:

$$\text{Perturbation} = \text{factor} * \eta, \ \eta \sim \mathcal{N}(0, \sigma^2), \tag{11}$$

$$\text{factor} = \begin{cases} 1, & \text{if } \sigma \leq 0.01 \\ 1.5 * (1 + e^{a-b\rho})^{-1}, & \text{otherwise}, \end{cases} \tag{12}$$

where $a = 90$ avoids early failure, $b = 0.125$ denotes adjusted rate of step simulation and $\rho$ is the number of steps past in the current episode. To enhance visual diversity, the four walls of the

Table 1: Baseline regression performance. Prediction lengths tested (second column) are $\{6, 12, 24\}$ for RP and $\{15, 30, 60\}$ for SimP. MAPE is on a $[0, 1]$ (not $[0\%, 100\%]$) scale. The best performance in each row is **bold**.

| Models | Autoformer | | DLinear | | TiDE | | TimeMixer | | TimeXer | | TimesNet | |
|---|---|---|---|---|---|---|---|---|---|---|---|---|
| Metrics | MSE | MAPE | MSE | MAPE | MSE | MAPE | MSE | MAPE | MSE | MAPE | MSE | MAPE |
| **RP** 6 | 28.40 | 1752 | 8.0e-3 | 17.75 | 2.6e-4 | 0.71 | 2.6e-3 | 5.17 | **2.5e-4** | **0.71** | 4.1e-4 | 1.23 |
| 12 | 53.01 | 2096 | 0.01 | 29.67 | 3.7e-4 | 0.97 | 5.6e-3 | 5.95 | **3.5e-4** | **0.90** | 5.0e-4 | 1.24 |
| 24 | 81.97 | 2420 | 0.01 | 27.39 | 6.0e-4 | 1.55 | 0.01 | 9.24 | **5.2e-4** | **1.34** | 7.6e-4 | 1.26 |
| **SimP** 15 | 1.00 | 403 | **1.0e-3** | **4.35** | 1.9e-3 | 5.38 | 3.7e-3 | 9.15 | 3.2e-3 | 8.05 | 3.5e-3 | 8.21 |
| 30 | 0.70 | 526 | **2.0e-3** | **9.69** | 2.2e-3 | 9.76 | 4.0e-3 | 22.96 | 3.5e-3 | 30.38 | 3.7e-3 | 29.21 |
| 60 | 0.70 | 3256 | 0.02 | 55.19 | **2.2e-3** | **30.78** | 3.6e-3 | 42.72 | 3.3e-3 | 40.81 | 3.5e-3 | 37.74 |

virtual room are each assigned a wall paper drawn randomly from a texture bank of 28 images, and 6 publicly available 3D furniture models[1] are positioned randomly in the robot's field of view. Falls are automatically detected depending on whether the z-coordinate of the robot's head is below a given threshold (set to 0.7 meters, slightly lower than the z-coordinate when the robot stands straight). All time-steps after the first such occurrence (if any) are labeled as falls.

## 4.2 REGRESSION TASK

We first experimented with a regression task (predicting future joint angles) since most TSF SOTA models are specialized for regression. The results demonstrate that such models do not even solve regression on our data, so adapting them to do classification is also unlikely to work.

Specifically, we evaluated 6 baselines cited earlier, using the implementations from Tsinghua University:[2] Autoformer, DLinear, TiDE, TimeMixer, TimeXer, and TimesNet. Their performance is shown in Table 1 using two metrics. The first metric, Mean Squared Error (MSE), is calculated after normalizing output data to a standard range, which facilitates peformance comparisons across datasets. However, it does not reveal the error relative to the actual values. Therefore, we also evaluate baselines using Mean Absolute Percentage Error (MAPE). Note that we report MAPE on a $[0, 1]$ rather than $[0\%, 100\%]$ scale and only TiDE and TimeXer can reduce the MAPE below 1 for small values of prediction horizon $P$. The poor regression performance suggests that most of these SOTA baselines would not be easily adaptable to our classification task. From these 6 baselines, we only retain TimesNet as a classification baseline, since its regression performance was relatively strong and it also includes a separate implementation branch for classification.

## 4.3 MAIN RESULTS

Our main experiment compared EMP with three classification baselines. In addition to TimesNet, our other baselines were FlowFormer because it is a SOTA TSF classification model and EgoFalls (Wang et al., 2023) because it is a model designed specifically for egovision-based fall detection. TimesNet and FlowFormer take joint motion as inputs, while EgoFalls is a vision model.

The historical sequence length $N$ is set to 24 and 60 for the RP and SimP datasets. We evaluate performances in the same range of prediction horizons, $P \in \{6, 12, 18, 24\}$ for RP and $P \in \{15, 30, 45, 60\}$ for SimP. Table 2 shows test accuracies (averaged over 5 independent training runs) for each model; these are also shown in Figure 3 for $P = 12$ on RP and 30 on SimP, since we finetune all hyperparameters on these configurations. For our best models on each dataset, Table 2 also reports average total training epochs with early stopping, and p-values for Walsh's t-tests with alternative hypotheses that our best models have higher average performance than each baseline. MP refers to an ablation of EMP without the egovision encoder.

Our approach consistently outperforms the other two baselines proposed to tackle the classification tasks in TSF in all cases. Our approaches (EMP or MP) yield at least 3.27% accuracy increment

---

[1]XWorld: https://github.com/PaddlePaddle/XWorld

[2]https://github.com/thuml/Time-Series-Library

Table 2: Results on the RP and SimP datasets with prediction lengths measured in time steps. For the RP dataset, we set prediction lengths to {6, 12, 18, 24} and {15, 30, 45, 60} for the SimP dataset. The best performance in each row is **bold** and the second best are underlined.

| Models | | Ours | | TimesNet | FlowFormer | Egofalls |
|---|---|---|---|---|---|---|
| | | EMP | MP | | | |
| RP | 6 | **70.91** | 67.27 | 70.91 | 70.00 | 54.55 |
| | 12 | **95.45** | 94.55 | 80.91 | 81.82 | 51.80 |
| | 18 | 89.00 | **93.00** | 81.00 | 87.00 | 42.50 |
| | 24 | **95.00** | 94.00 | 86.00 | 89.00 | 50.00 |
| Training epochs | | 8.23 | 8.03 | - | - | - |
| T-test $p$-value for $P = 12$ | | - | - | 5.2e-5 | 2.0e-4 | 7.3e-24 |
| SimP | 15 | **96.60** | 96.00 | 79.90 | 90.90 | 78.17 |
| | 30 | **95.90** | 90.40 | 74.10 | 80.00 | 76.35 |
| | 45 | **96.10** | 89.50 | 76.70 | 76.10 | 78.08 |
| | 60 | **95.50** | 87.90 | 75.80 | 76.30 | 79.65 |
| Training epochs | | 8.53 | 9.90 | - | - | - |
| T-test $p$-value for $P = 30$ | | - | - | 0.024 | 2.3e-18 | 1.5e-40 |

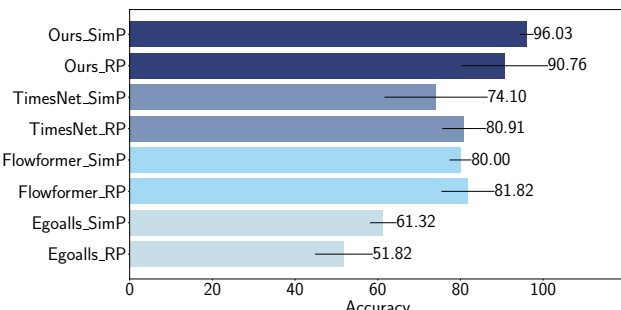

Figure 3: Classification accuracy for $P = 12$ (real data, RP) and $P = 30$. Error bars show standard deviation over 5 experimental repetitions.

(67.27% against 64.00%) on RP and 5.10% (96.00% against 90.90%) on SimP. The early stopping mechanism reduces training epochs (8.23 against 8.53 and 8.03 against 9.90) to avoid overfitting the smaller RP dataset. The results show that EMP is better suited to our task than existing SOTA models, possibly because they are designed using benchmark data with more periodicity.

In most cases, MP (without visual input) is also able to surpass the SOTA models by a large margin. In fact, MP is closer to previous models in that it is unimodal. This suggests that a primary factor in EMP's better performance is the use of planned trajectories as exogenous variables, as opposed to more artificial exogenous variables such as time stamps. Counterintuitively, MP even outperformed EMP on the RP dataset when $P = 18$ (93.00% against 89.00%), suggesting that visual cues might be more susceptible to overfitting. We conducted another 25 repetitions to probe this effect, and the performance gap reduced from 4% to 1.67% (MP is still better), but a t-test showed that the difference between MP and EMP was not statistically significant.

## 4.4 ABLATION STUDIES

We conduct ablation studies on each encoder and report the accuracies, training time (in seconds per epoch) and t-test between EMP and other design accuracies in Table 3. The prediction length is set to 12 and 30 for the RP and SimP datasets, respectively. The base design (P) leverages proprioception (i.e., observed joint angles) only and the ultimate design (EMP) leverages all modalities and

Table 3: Encoder ablation results. "E," "M," and "P" indicate whether the **E**gocentric vision encoder, the **M**otion planner trajectory encoder, and the **P**roprioceptive joint sensor motion encoder are included. 30 repetitions for each design. Best performance in bold.

| Design | | EMP | MP | EP | P |
|---|---|---|---|---|---|
| | Acc | **95.45** | 94.55 | 92.73 | 89.09 |
| RP | Time | 20.84 | 16.52 | 17.48 | **16.29** |
| | T-test $p$-value | - | 0.22 | 0.91 | 0.33 |
| | Acc | **95.90** | 90.40 | 86.60 | 85.90 |
| SimP | Time | 1.58 | 1.61 | 1.37 | **1.17** |
| | T-test $p$-value | - | 2.9e-12 | 2.0e-13 | 3.7e-11 |

Table 4: Input data ablations. 5 repetitions for each design. Best performance in bold.

| Design | | EMP | Real position | Historical only |
|---|---|---|---|---|
| RP | Acc | **95.45** | 94.55 | 94.54 |
| | T-test $p$-value | - | 0.79 | 0.79 |
| SimP | Acc | **95.90** | 84.80 | 84.20 |
| | T-test $p$-value | - | 1.4e-4 | 9.0e-4 |

variables. The vision modality and exogenous variables do not incur much more training time, but combining more modalities improves performance particularly on the larger SimP dataset.

In Table 4, we investigate the importance of planned future trajectories as exogenous model input. Our ultimate model (EMP) takes the *planned* trajectory for the past $N$ time-steps and future $P$ time-steps, a matrix in $\mathbb{R}^{(N+P) \times C}$. Note that the *actual* joint angles in the future are not observable and should not included, but in general the *planned* angles for the future will be available, because they are generated in advance by a motion planner and drive the robot's position control. We test two ablations of the input data provided to the trajectory encoder: **(1)** instead of the full $N + P$ timesteps, we only supply the past $N$ time-steps of planned joint angles ("historical only"), and **(2)** instead of the past $N$ planned joint angles, we provide the past $N$ *actual* joint angles, i.e., a copy of the input given to the motion encoder ("real position"). The ablation effects are not statistically significant on RP, but on SimP, the difference between real position and historical only (84.80 against 84.20) is negligible compared to that between EMP and historical only (95.90 against 84.20). This shows that providing exogenous variable values in the future (when justified, as is the case for motion trajectories planned in advance) can enhance performance, especially on SimP. The smaller differences on RP may be due to more limited diversity in the motion plans. Compared with simulation, it is less feasible on real hardware to test a broad range of trajectories when many may cause a fall.

## 5 CONCLUSION AND FUTURE WORK

In this paper, we first observe that most existing TSF benchmarks are limited to periodic data, and so TSF models trained on those benchmarks may not work well for aperiodic data. Second, we distinguish two kinds of classification tasks in TSF by their temporal labeling granularity. In some real world applications, forecasting the future per-timestep class labels could be more informative than assigning a single label to an entire time-series. Third, due to the lack of proper datasets addressing the foregoing issues, we contribute our own benchmark datasets from a robotics application, one real and one simulated. The datasets are freely accessible for the research community. Lastly, we show that EMP is a simple yet effective model for our benchmark, leveraging both visual and proprioception data, rather than employing or finetuning existing vision models on time series data. EMP outperforms previous TSF SOTA models and hence establishes a strong new baseline for the new challenges in our benchmark dataset. We also noticed that the aperiodic data is still limited especially when exogenous variables highly depend on scenarios and other simple yet effective layers are left unexplored, which may further simplify our model and generalize it to different domains.

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

## A  IMPLEMENTATION DETAILS AND HYPERPARAMETERS

All models are trained and tested on one single NVIDIA A40 GPU with 5 repetitions in each condition. The Adam (Kingma & Ba, 2014) algorithm is used for optimization. We use up to 10 training epochs with early stopping, using a "patience" parameter of 3 epochs, and batch size of 4, which worked best for time-effective data loading (dominated by the visual input data). Detailed hyperparameters for our ultimate model, EMP, are listed in Table 5.

## B  NORMALIZATION LAYERS

We further conduct extensive experiments on other widely used normalization layers and report the performance in Table 6. Normalization over spatial dimension consistently achieves the highest accuracy across all designs, demonstrating its effectiveness.

## C  VISUALIZATION

Walking requires joint coordination but not all joints are equally important in this process. A reasonable model should be able to capture the key joints. To better understand how the model works on this task, we visualize the weights of the projection layer on both datasets. There is a clear pattern in the plot for the RP dataset, where only a few horizontal lines contains the darkest or brightest color indicating the walking process is highly related to several joints rather than all of them. This observation fits our motivation for the motion encoder. We still can find the bright and dark lines

Table 5: The hyperparameters for our model on datasets RP and SimP. A merged cell indicates the hyperparameter is the same for both datasets.

| Datasets | | RP | SimP |
|---|---|---|---|
| | learning rate | 0.0005 | 0.0001 |
| | weight decay | 0.0001 | 0.00001 |
| Motion module | num_Templayers | 6 | |
| | num_Spatlayers | 2 | |
| | normalization | Normalization over spatial dimension | |
| | learning rate | 0.0001 | 0.001 |
| | weight decay | 0.0001 | 0.0001 |
| Trajectory module | num_Timeslayers | 2 | |
| | inc_dim | 256 | |
| | top_k | 3 | |
| | learning rate | 0.00001 | 0.000001 |
| | weight decay | 0.0005 | 0.000001 |
| Egovision module | num_convlayers | 4 | |
| | fft_threshold | 30 | |
| | learning rate | 0.0001 | 0.001 |
| Classifier | weight decay | 0.0001 | 0.0001 |
| | d_model | 1024 | |

Table 6: Effectiveness of different normalization. BatchNorm stands for the 1D batch normalization.

| Normalization | | Ours (Spatial) | LayerNorm | BatchNorm. |
|---|---|---|---|---|
| | P | **89.90** | 70.45 | 72.88 |
| RP | EP | **92.73** | 62.88 | 84.09 |
| | MP | **94.55** | 73.48 | 79.24 |
| | EMP | **95.45** | 71.67 | 80.45 |
| | P | **85.90** | 71.37 | 60.68 |
| SimP | EP | **86.60** | 76.17 | 74.45 |
| | MP | **90.40** | 75.67 | 74.62 |
| | EMP | **95.90** | 80.48 | 80.13 |

in Figure 4b, even though the pattern is less salient for the SimP dataset. A reason for such difference is that the SimP perturbations are introduced into the trajectory evenly over joints, which might decrease the distance between them in the latent space.

More comprehensive autocorrelation plots for public datasets and our datasets are shown in Figure 5. Most TSF datasets contain explicit periodicity except for the exchange rate and ours.

The three locations on campus for testing our robot, Poppy Humanoid, are shown in Figure 6a, Figure 6b and Figure 6c.

The final robot poses are depicted in Figure 7 from the bird-eye view for each episode in SimP. More specifically, the horizontal and vertical axis refers to the X- and Y-coordinates of Poppy's head at the last timestep. The falling episodes (red crosses) form a circle and are further from the original point, assuming start coordinate is always $(0, 0)$, compared to the non-fall episodes (green dots). The reason is that our robot can walk along all possible directions and will eventually fall in a falling episode, in which it is lying on the ground instead of standing on the ground with a lower Z-coordinate but greater (in magnitude) X- and Y-coordinates of its head. Note that our models do not take X- or Y-coordinates as inputs and therefore cannot take advantage of any clear borders readers may find in Figure 7 in the learning phase.

It is also crucial to show that our models are proposed for tackling forecasting problems rather than as a persistence model. We randomly select 2 falling episodes and plot the Z-coordinates of Poppy's

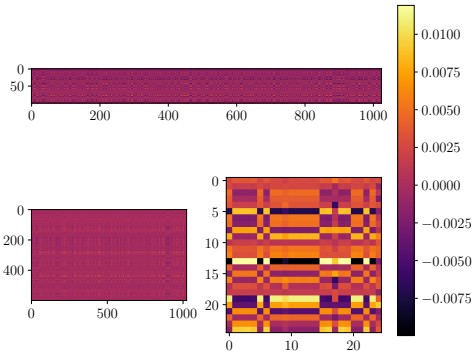 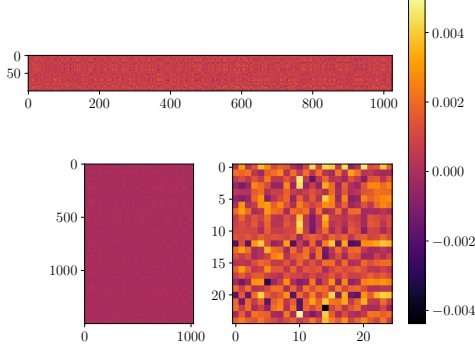

(a) The motion encoder trained on the RP dataset.

(b) The motion encoder trained on the SimP dataset.

Figure 4: Visualization of weights ($N * C$) in the motion encoder. The vertical axis refers to channels in the historical sequence while the horizontal axis refers to the output channels of the projection layer. The original weight matrix (bottom left) is zoomed in twice by truncating the size of input channels (upper) and the size of both input and output channels (bottom right).

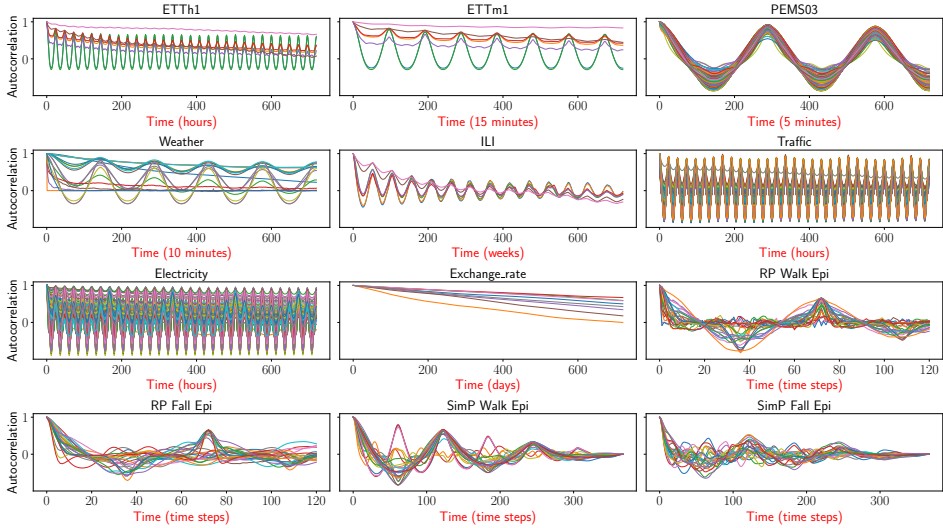

Figure 5: Autocorrelation plots for standard TSF benchmark datasets as well as our datasets (RP and SimP, details in the text). Autocorrelations near 1 indicate periodic signals. Each time step is roughly 100 ms long for our RP dataset and 33 ms for our SimP dataset.

head in Figure 9. The vertical breaking lines refer to the latest *possible* frame for sampling history sequences from an episode, given different prediction spans, $[15, 30, 45, 60]$. They are regarded as the stop sign for data sampling, which guarantees that the last frame of history sequences cannot exceed the corresponding vertical line. In other words, the sliding window of an input history sequence is always on the left side of the vertical line and no falling signals are included in the window. Therefore, we are confident to say our model is truly forecasting the onset of a fall.

## D SUPPLEMENTARIES

The definition of MAPE is given as:

$$\text{MAPE} = \frac{|y - \hat{y}|}{y} \times 100\%, \tag{13}$$

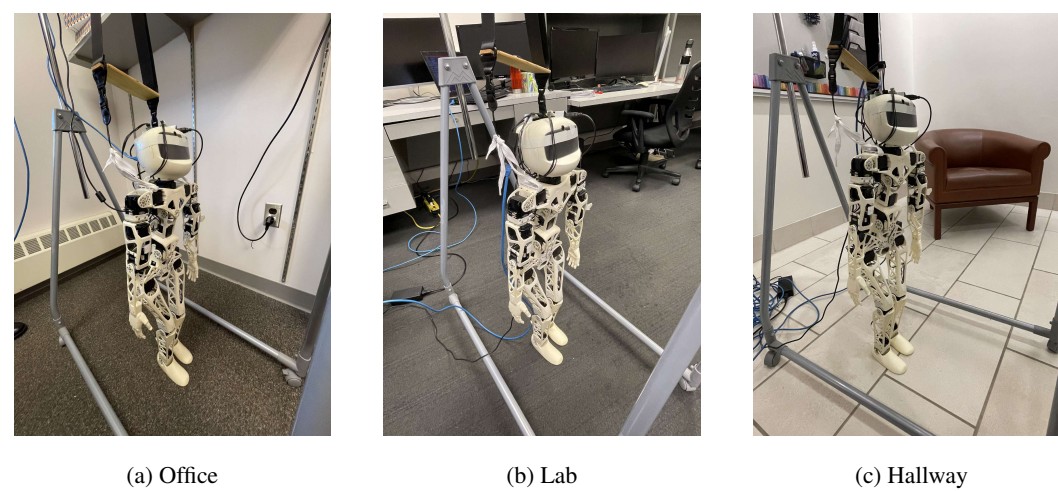

(a) Office           (b) Lab           (c) Hallway

Figure 6: The three locations where the data for the RP dataset is collected.

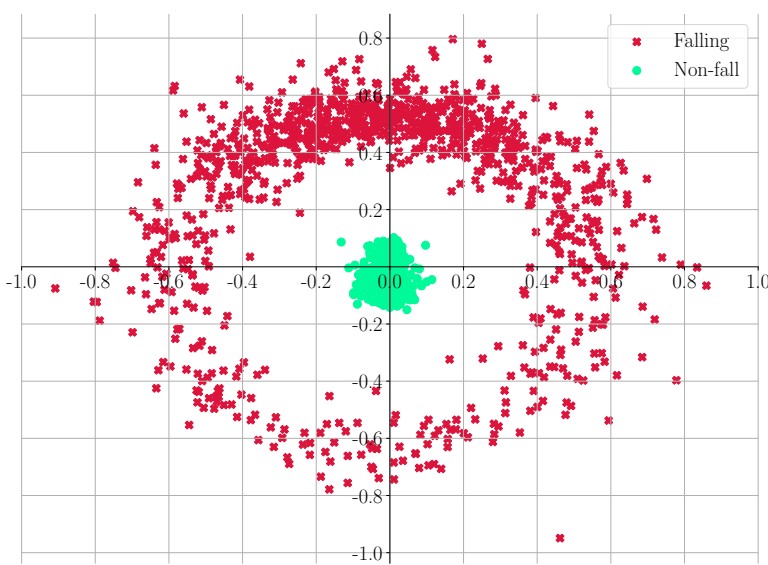

Figure 7: Destination distribution of our SimP dataset in the XY plane.

where $y$ and $\hat{y}$ stand for the ground truth and prediction given by baselines.

## E  NON-DEEP BASELINES

We implemented 2 non-deep baselines, logistic regression and linearSVC, and show their performance on our datasets in Table 7. Shallow baselines are competitive only in short prediction spans on the RP dataset. Our model outperforms the shallow baselines in long-term prediction even after dropping the visual clues (i.e. the MP ablation). Our supplementary experiments also show that shallow baselines require dedicated engineered features, which are all dependent on expertise and experience, while our deep models can take care of this by simply taking raw data as inputs.

A representative frame from RP  A representative frame from SimP

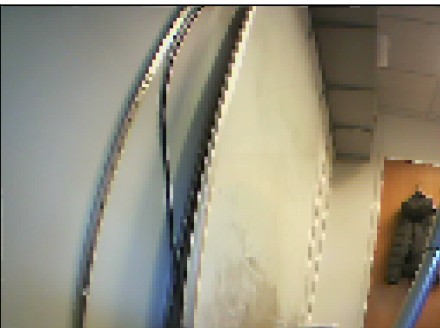 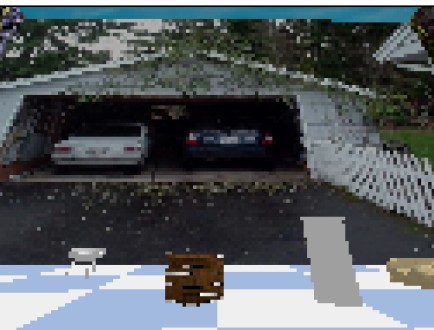

Figure 8: Representative frames for our RP (left) and SimP (right) datasets. The furniture shown in the right figure includes a stove, barrel, bookshelf and bed (from left to right). Furniture is randomly placed for each episode to partially obstruct Poppy's view.

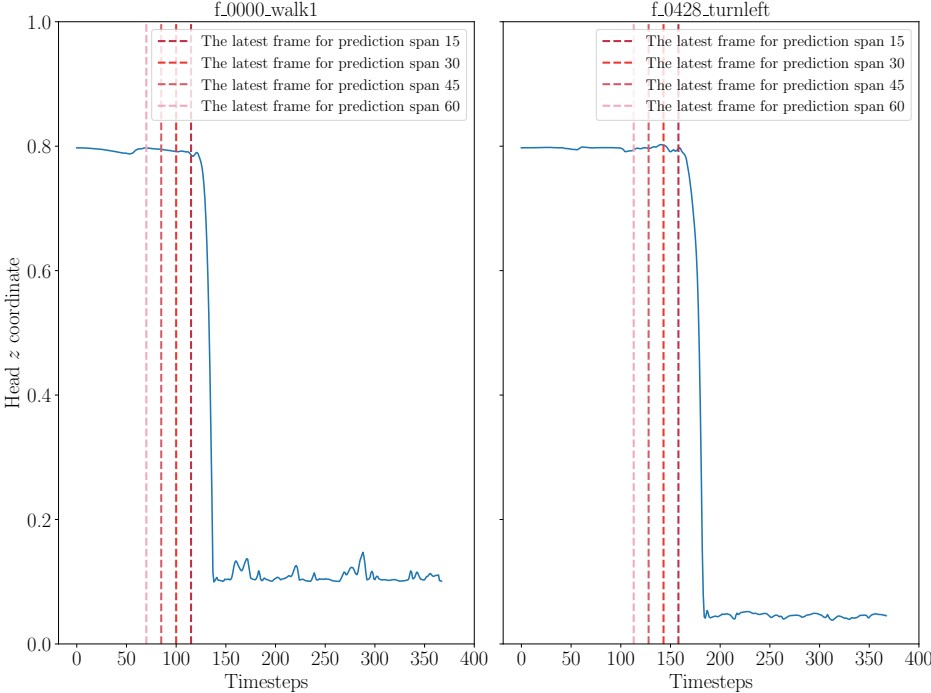

Figure 9: The Z-coordinates of Poppy's head and the stop line for data sampling.

Another interesting finding is that our results verify shallow baselines' effectiveness on a low rate of interest (ROI, ROI = best prediction span/history sequence length). This metric measures the inference ability of a given model. The prePARE (Dey et al., 2022) paper reported a history sequence of 3.6s and reached the best performance with prediction span of 0.72s, which indicates an ROI of $0.72/3.6 = 0.2$. On our dataset, the shallow baselines we tested also exhibit a decent performance at prediction span 6 indicating an ROI of $6/24 = 0.25$. But, only our EMP model can generalize to longer prediction spans.

Table 7: Results of non-deep models on the RP and SimP datasets with the same configurations in Table 2.

| Models | | Ours | | Logistic regression | | LinearSVC | |
|---|---|---|---|---|---|---|---|
| | | EMP | MP | MP | P | MP | P |
| RP | 6 | 70.91 | 67.27 | **81.82** | 80.91 | 81.82 | 81.82 |
| | 12 | **95.45** | 94.55 | 80.00 | 82.73 | 79.10 | 84.85 |
| | 18 | 89.00 | **93.00** | 81.00 | 84.00 | 83.00 | 87.00 |
| | 24 | **95.00** | 94.00 | 75.00 | 80.00 | 76.00 | 81.00 |
| Training epochs | | 8.23 | 8.03 | - | - | - | - |
| SimP | 15 | **96.60** | 96.00 | 75.60 | 74.10 | 73.90 | 73.10 |
| | 30 | **95.90** | 90.40 | 71.10 | 72.20 | 69.80 | 69.70 |
| | 45 | **96.10** | 89.50 | 70.10 | 74.00 | 69.70 | 74.70 |
| | 60 | **95.50** | 87.90 | 70.20 | 73.90 | 69.60 | 73.00 |
| Training epochs | | 8.53 | 9.90 | - | - | - | - |

## F  MULTIMODAL ENSEMBLES OF DEEP BASELINES

Due to the lack of open source multimodal baselines in the TSF field, we use an ensemble of the three baselines (TimesNet, FlowFormer and Egofalls). Since Egofalls takes visual input whereas the others can be used on joint angle input, this ensemble effectively becomes a multimodal predictor, for potentially fairer comparison with EMP. The final prediction of this ensemble is determined by majority voting among the three baselines.

We investigate three forms of this ensemble: **(1)** using historical joint proprioception (P) as input for both TimesNet and FlowFormer, **(2)** using historical data (P) as input to FlowFormer and planned motion trajectories (M) as input to TimesNet, or **(3)** using historical data (P) as input to TimesNet and planned motion trajectories (M) as input to FlowFormer. Table 8 shows the results for each ensemble.

EMP can still outperform all ensembles in most cases, especially on all long prediction spans. The one counterexample is on RP with prediction span 6 on our RP dataset reports an average accuracy higher than individual models used in the ensemble (72.73% against 70.91%, 70.00% and 54.55%, see Table 2). This observation indicates that individual baselines are more likely to give a correct prediction when forecasting a nearer future than foresee a state at further timestep. This indicates that the multimodal baseline ensemble may be competitive for short-term prediction on a small dataset, but EMP is generally more effective for longer prediction spans.

Table 8: Prediction accuracy of multimodal ensembles of the three deep baselines (TimesNet, Flow-Former and Egofalls) on the RP and SimP datasets.

| Models | | EMP (ours) | Ensemble 1 | Ensemble 2 | Ensemble 3 |
|---|---|---|---|---|---|
| RP | 6 | 70.91 | **72.73** | 66.36 | 66.36 |
| | 12 | **95.45** | 81.82 | 66.36 | 70.00 |
| | 18 | **89.00** | 78.00 | 71.00 | 68.00 |
| | 24 | **95.00** | 88.00 | 71.00 | 69.00 |
| SimP | 15 | **96.60** | 89.10 | 91.90 | 92.00 |
| | 30 | **95.90** | 82.60 | 88.90 | 90.50 |
| | 45 | **96.10** | 77.80 | 88.80 | 90.50 |
| | 60 | **95.50** | 79.00 | 85.90 | 89.40 |

