# OpenReview forum: "A Multimodal Label Forecasting Method for Aperiodic Visuo-Motor Time Series"
_ICLR.cc/2026/Conference — Submitted to ICLR 2026_

### Official Review · Reviewer_uCay · 2025-10-27

**Soundness:** 2
**Presentation:** 3
**Contribution:** 3
**Rating:** 4
**Confidence:** 4

**Summary:**

This submission presents a new benchmark and method for Time Series Forecasting (TSF). It argues that the community is overly focused on periodic regression tasks and targets a new fine-grained classification challenge: predicting per-time-step fall labels for a humanoid robot using aperiodic, multimodal sensory input.

To this end, it presents two valuable new datasets (RP: real-world, SimP: simulated) containing egocentric video, proprioceptive data (actual joint angles), and planned motion trajectories. In addition, a EMP model is proposed which uses three parallel encoders to process this data. A key feature of this model is its use of the planned trajectory as an exogenous variable, including the portion of the plan that extends into the future ($P$ steps).

Experiments show that EMP (even the ablated version MP) outperforms several TSF models like TimesNet and FlowFormer and the domain-specific EgoFalls on this new benchmark. The authors conclude that EMP is a superior baseline for this new problem, and they attributes EMP’s success to its multimodal architecture and designs for handling aperiodic data.

**Strengths:**

**(S1)** Clear problem formulation and new benchmarks. This work provides two new datasets (RP and SimP) for TSF classification under aperiodic, real-world robotic conditions. This fills a substantial gap in existing TSF benchmarks that focus on periodic or stationary signals. The new task of per-time-step fall prediction is of clear practical and research importance.

**(S2)** The use of exogenous data. To my knowledge, the key contribution of this work is its use of the planned motion trajectory as an exogenous variable. Leveraging the known future $N+P$ steps of this plan is a fresh and powerful advance. This is a far more meaningful use of exogenous data than the field's standard (and rather weak) reliance on simple time-stamp encodings.

**(S3)** Thorough experiments. The empirical work is robust. It first shows that SOTA regression models fail on this data in Tab. 1, justifying their new method. The comparison with relevant classification and domain-specific baselines shows big improvements with significant p-values in both simulated and real robot scenarios. What I appreciate the most are the multiple repetitions, standard deviations, and statistical tests (Walsh's t-tests and p-values), which are applied to support claims of significance, rather than just cherry-picked runs.

**(S4)** Ablation studies and analysis. Tab. 3 and Tab. 4 provide ablation results to break down the contributions of each encoder (vision, proprioception, and trajectory) and the significance of future planned trajectories especially in simulation. Particularly, MP ablation (removing vision) reveals that the model's strength derives mainly from the proprioceptive and trajectory inputs, not the vision stream. Training overhead for added modalities is also reported for efficiency evaluation.

**Weaknesses:**

**(W1)** This work’s core premise seems to be directly contradicted by its method. It is built on the premise of tackling aperiodic data. However, in Line 272, the Trajectory Encoder is designed specifically to exploit the "more reliable periodicity... in a repetitive gait" of the planned trajectories, and uses an FFT-based filter to do so. This is a direct contradiction. I recommend the authors add clarifications in the revised manuscript to address this point.

**(W2)** IMHO, the comparison seems a bit unfair. The baselines were almost certainly not given access to the same powerful exogenous data (the future plan) that the EMP model uses. In other words, the MP ablation (without vision) achieves 90.40% on SimP, while the best baseline (FlowFormer) scores only 80.00%. This may suggest the 10.4% gap is not due to EMP architecture, but rather that MP has access to $N+P$ future plan and baselines do not. I look forward to the author's response on this and hope they can incorporate insightful comments into the revision.

**(W3)** An unexplained "sim-to-real" gap exists. Key components (like the future plan and the vision encoder) are critical in simulation but are marginal or even detrimental on the real-world data. Specifically, the future plan is essential on SimP (with an 11.7% drop when removed) but irrelevant on RP (only a 0.01% drop). The explanation now (“limited diversity”) is a bit insufficient.

**(W4)** The task definition is a bit ambiguous. It is unclear whether the model is truly forecasting the onset of a fall or simply learning a trivial persistence model (i.e., "if I am already falling, I will still be falling in $P$ steps"). In this case, it is not a forecasting task. I believe this needs further clarification.

**(W5)** Several limitations in RP dataset. The RP dataset is limited in size (110 episodes) and diversity of environments and trajectories. Also, there are not enough scenarios for each environment (only three locations), which could affect the validation.

**(W6)** The title of Sec. 2.3 is too general. Since it is specifically about multimodal time series forecasting, revising the title to “Multimodal TSF” would better reflect its scope and avoid confusion with general multimodal representation learning.

**(W7)** Some notation issues reduce readability. (i) The symbol **M** (in Line 156, Sec. 3.2.1) and subsequent uses are overly bolded, which is a bit wired in the entire manuscript. (ii) Similarly, **V**, **K**, and **T** in Eqs. 8–10 suffer from the same issue. (iii) Some operators and function names are typeset in italics, which are commonly noted as upright (\text{} or \operatorname{}) to distinguish them from variables. Specifically, (e.g., “Spa()” and “TepMLP()” in Eq. 6, “Conv3D” in Eq. 9, “ConvBlk” in Eq. 10, “Perturbation” and “factor” in Eqs. 11–12).

**Questions:**

Most of my concerns and related recommendations have been stated in the Weaknesses section. I encourage the authors to focus their efforts on addressing those points, as they are critical for strengthening the manuscript in the rebuttal stage.

The following are more specific, minor questions to help the authors think more deeply about certain design choices and experiment setups, which I hope might be helpful for this and future work:

**(Q1)** Given that ablation shows unimodal MP often matches or outperforms the multimodal EMP (especially for certain $P$ values on RP), what regularization strategies were attempted, if any, to stabilize multimodal learning? What insights can the authors offer about the situations/environments where fusion of vision actually delivers benefit?

**(Q2)** Would EMP's architecture (vision + proprioception + exogenous planning) port well to different domains, or is the value mainly restricted to robotic forecast settings where planned forward trajectories are known and the main source of exogenous data?

**(Q3)** Have variants where DCT/iDCT is replaced with learned temporal filters like Conv kernels been tried? How significant is the DCT feature transformation compared to other modeling components?

---

## Justifications:

The main idea of this paper is fresh and strong. The new datasets and the focus on exogenous future plans are great advances, and the authors have clearly identified a valuable problem. More importantly, I can catch the soundness of the experiments and analysis.

However, several concerns exist as listed in Weaknesses. The model uses future trajectory plans as input, while baselines appear not to. The difference in performance is probably due to unequal information, not better design. Additional concerns include the contradiction between the aperiodic framing and the method’s reliance on periodic gait, the sim-to-real gap, and whether the task truly involves forecasting or label persistence.

Therefore, I cannot recommend acceptance at this stage and give a rating of 4. I would be glad to raise my rating if thoughtful responses and improvements are provided in the rebuttal stage. I am also open to follow-up discussions with the authors to help further strengthen this work.

I hope these comments help my fellow reviewers and ACs understand the basis of my recommendation.

---

> ### Author Response · Authors · 2025-11-21
>
> Thank you very much for your thorough and helpful review, and your openness to increasing your rating.  This is one of the highest-quality reviews we have received at recent top conferences.  Here are our responses to the weaknesses and questions:
> * (W1) This point is well-taken, but we would like to point out we assume periodicity *only in the trajectory encoder*, since the motion plan can be periodic, whereas the actual observed joint positions, visual input, and importantly the fall/stand labels, are not periodic.  We will add this language near line 272 to further emphasize the very limited periodicity in our dataset.  FFT is also used in the vision encoder but that is to low-pass filter each image frame, and does not operate along the time dimension.
> * (W2) This is a fair point.  We will add text in section 4.3 explaining that the comparison is not completely fair because no existing baselines were perfectly suited to our dataset, but these were the closets we could find.  We will also clarify in the text exactly which modalities were provided to which baselines (joint motion and planned trajectories only to TimesNet and FlowFormer, vision only to Egofalls).  A potentially fairer comparison would be to use a mixture model/ensemble of these three baselines, which effectively becomes a multimodal predictor.  We are working to complete this experiment before the rebuttal window closes.
> * (W3) We will add more detail to the paper that emphasizes the substantial differences in motion diversity between RP and SimP.  When we calculate the variance (across episodes) of each joint angle at each time-step, and then average those variances across joint angles and time-steps, the result is over 10x higher on SimP than RP.  Consequently, the motion plans in SimP (including non-fall ones) produce a wide range of final orientations and positions of the robot, whereas the RP motion plans mainly walk forward in a straight line.  Given these details, does the reviewer agree the highly diverse motions in SimP are a reasonable explanation for the higher importance of multiple input modalities?   We will add to the paper the 10x variance ratio and a scatter plot from birds-eye view of final robot poses in SimP to highlight this distinction. Please refer to the new Figure 7 and our explanation in Appendix C.
> * (W4)  This is an excellent question, and we did take special care to ensure the model was not simply a persistence model.  In particular, in SimP, we use the head $z$ coordinate to detect fall onset, and ensure that the prediction window starts before a fall and (in the case of fall examples) ends after a fall.  We test prediction spans up to 1 second which can comfortably include a pre-fall period.  We will add these details to our appendix along with a plot showing representative head $z$ coordinate time-series and where the prediction span time range sits along those time series. Please refer to the new Figure 8 and our explanation in Appendix C.
> * (W5) We agree the RP size is a significant limitation, but given the time-intensive nature of physical robot data collection we may not be able to address this before the rebuttal window closes.  We will add a sentence to the conclusion emphasizing the importance of more data collection in future work.
> * (W6) Point taken, we will change the title of 2.3 as suggested.
> * (W7) For points (i) and (ii), we agree the formatting looks a bit odd, but in fact this is ICLR’s official formatting recommendation for tensors.  For (iii), we have de-italicized these terms as suggested.
> * (Q1) As explained above for (W3), our hypothesis is that motion diversity is a main driver for the utility of visual input modality.
> * (Q2) In principle, EMP’s motion and trajectory encoders can work with any numeric time series arranged in NxT arrays, so there is potential it might work on other non-robotic data that combines video and lower-dimensional numerical measurement vectors.  Of course, that hypothesis is not addressed by our current experiments.  We will add this as another direction for future work in our conclusion.
> * (Q3) DCT/iDCT layers are proven effective in 3D human pose short prediction [1]. This comparison was between a MLP with DCT layers and RNN, CNN and GCN models rather than at layer-level. This is an interesting question that would make for an informative ablation experiment. If time permits, we will try this before the rebuttal window closes.
>
> [1] Guo, Wen, et al. "Back to MLP: A simple baseline for human motion prediction." Proceedings of the IEEE/CVF winter conference on applications of computer vision. 2023.

---

> ### Comment · Reviewer_uCay · 2025-11-23
> **Official Response by Reviewer uCay**
>
> To the authors,
>
> I would like to thank the authors for their effort put into running additional experiments during the short rebuttal period. Most of my concerns have been addressed after going through the authors' response and the other reviewers' comments.
>
> Specifically:
> - W1 (Periodicity contradiction): I accept the distinction between the planned trajectory (which holds periodicity/repetitive gait) and the realized observation (aperiodic due to noise/falls). The clarification that the FFT in the vision encoder acts as a low-pass filter rather than a periodicity extractor is also helpful.
>
> - W3 (Sim-to-Real / Vision utility): The analysis regarding the motion variance (10x higher in SimP) and trajectory diversity visualization provides a stronger justification for why vision is crucial in simulation but marginal in the current real-world dataset.
>
> - W4 (Forecasting vs. Persistence): The explanation of the sampling strategy relative to the head Z-coordinate convinces me that the model is performing genuine forecasting rather than trivial state persistence.
>
> Also about Reviewer wuXv’s point, the simple baselines (Logistic Regression/LinearSVC) in Appendix E is a valuable addition, which shows learning methods is specifically necessary for long-term horizons.
>
> Give that the authors have addressed the concerns regarding aperiodic framing and the utility of vision, I raise my rating from (4) to (6). I hope that all these clarifications, promised modifications, and new experiments would be integrated into the revised manuscript **with highlighted text color** to show your improvements. It seems now we still cannot see the revision. This would largely improve the overall quality this work, especially for the (W2) concerns. I also welcome further discussion with the authors (if the rebuttal period permits) and am happy to provide suggestions to help further improve this work.
>
> Best,
>
> Reviewer uCay

---

> > ### Author Response · Authors · 2025-11-23
> >
> > Thank you for your quick response.  We have uploaded a second revision, same as the previous one except all changes so far are now indicated with red text color.  Some changes we promised are still in progress and we will upload them in another revision as soon as possible.  Right now, we can see our changes reflected in the PDF when we click the PDF download icon at the top right of this page.  Could the reviewer please confirm that when they click this icon, they see changes so far including red text, new appendix section E, figures 7-8, and table 7?  We will post one more revision with the remaining changes we promised once our experiment to address W2 is finished.

---

> > > ### Comment · Reviewer_uCay · 2025-11-26
> > > **Official Response by Reviewer uCay for Revision Checking**
> > >
> > > To the authors,
> > >
> > > Thanks for the timely update. I can confirm that I can clearly see the new contents marked in red text in the revised Appendix. I appreciate these updates and look forward to your further revisions. In addition, I recommend writing a global response after completing all the updates, listing all significant clarifications, modifications, and updates one by one. This would help other reviewers and ACs understand your improvements and the post-rebuttal quality of this work.
> > >
> > > Best regards,
> > >
> > > Reviewer uCay

---

> > > > ### Comment · Reviewer_uCay · 2025-11-26
> > > > **Official Response by Reviewer uCay for Revision Checking (Part 2)**
> > > >
> > > > Plus, if you take this suggestion, please consider explicitly noting if you think each reviewers' concerns have been addressed with evidence in this global response (even if the reviewer does not reply). IMHO, this would be helpful for all reviewers and ACs grasp the full picture smoothly.

---

### Official Review · Reviewer_p6gi · 2025-10-29

**Soundness:** 2
**Presentation:** 2
**Contribution:** 2
**Rating:** 2
**Confidence:** 4

**Summary:**

This paper focuses on the task of predicting time series that lack periodicity. The specific contribution lies in providing both a real and a simulated dataset to determine whether the robot has fallen. Each frame in the dataset has its own label for each time step. Additionally, a planned joint trajectory is used as the exogenous time series to assist in label prediction.

**Strengths:**

- The paper is clearly presented

**Weaknesses:**

The issue of non-periodic label prediction at each time step emphasized in this work is essentially a time series classification or anomaly detection problem. This is common in general time series classification or anomaly detection tasks, as seen in the widely used UCR database, which contains many data samples from the robotics application field. Therefore, from this perspective, the novelty of this work does not seem significant to me. Specifically, the label forecasting mentioned in the title can more accurately be described as classification.

Additional suggestions:

- The comparative methods could be thoroughly compared with multimodal methods related to time-vlm.
- This work also seems relevant to multimodal action recognition or motion prediction, and there should be many related studies in the field of computer vision.
- The term $𝐿_{t+P}$  in line 134 is ambiguous. It is unclear whether it refers to a single value or P values, since you mentioned "occurring P time steps in the future".
- There is too little information regarding vision sequences in the text.
- More visual analyses should be included, discussing the challenges of the classification tasks studied (beyond the so-called non-periodicity).

**Questions:**

- What are the vision sequence inputs you meant in the model figure?

---

> ### Author Response · Authors · 2025-11-21
>
> Thank you for your constructive suggestions.  Regarding “classification” vs. “forecasting,” we feel there is an important distinction here.  Many time-series “classification” tasks include the event being labeled as part of the observed data.  For example, in action recognition, the input video would *include* the occurrence of the fall, which the model detects when it labels the video as a fall.  In contrast, “label forecasting” means the relevant action occurs after the end of the observed input data to the model.  In some sense, label forecasting is strictly harder than recognition, since any label forecasting model can be repurposed for recognition (just trim the input video), while the reverse is not true.
>
> Furthermore, we only found two robotics-related dataset (SonyAIBORobot Surface andSonyAIBORobot Surface II) in the UCR repository and they are substantially smaller (adds up to a total of 47 examples for training) than the datasets we collected.
>
> Regarding the additional suggestions:
> * We did not compare with time-vlm because it is not a true multimodal model.  It takes one raw input stream (a time-series of numeric measurements) and then preprocesses it in multiple ways for different model pathways (e.g., the visual pathway essentially ingests a visual plot of the input stream).  This is not the same as multiple raw input streams (joint angles and egovision) so it can not be used directly with our multimodal dataset.
> * As mentioned above, we consider action recognition a distinct and easier task than forecasting.  It is surprisingly difficult to find models that meet all the criteria of our datasets (multimodal, egocentric visual stream, robot with joint angle data rather than human).  The closest we could find is Egofalls, which we already did include (Table 2) to address this kind of point.  We already discuss other related computer vision work in section 2.4.
> * We will add “where $P\in\mathbb{N}$” at the end of this sentence to clarify that $P$ is a single value (we use different values of $P$ in different experimental configurations, but in each individual experiment $P$ is a single number).
> * The vision sequences are batches of videos of RGB frames, recorded from the robot’s head camera, as we already describe in section 3.2.2.  This section also discusses some challenges (the robot body is mostly out of view, falls depend more on relative movement/optical flow than objects in view).  If space permits, one thing we will do is add a figure with some representative frames from the input videos to make it more clear.  If there are other specific challenges or pieces of information the reviewer wanted to see here, could they please let us know what those are, so we can add them?

---

> ### Author Response · Authors · 2025-12-02
>
> We would like to point out another infeasibility. The two sub-datasets mentioned above are uni-modal. It is infeasible to test our multimodal model on unimodal datasets.

---

### Official Review · Reviewer_bxW8 · 2025-11-01

**Soundness:** 3
**Presentation:** 3
**Contribution:** 3
**Rating:** 6
**Confidence:** 2

**Summary:**

This paper introduces a new robotics benchmark for aperiodic time series forecasting (TSF): predicting falls in humanoid locomotion using egocentric vision and proprioception. The authors contribute two datasets (real hardware “RP” and simulated “SimP”)  from a humanoid robot. They also propose a multimodal method that leverages observed states, planned trajectories (exogenous inputs), and vision for per-timestep classification, showing improvements over existing TSF baselines.

**Strengths:**

Overall, the paper is well-written and easy to follow. Given the lack of aperiodic benchmarks, the two collected datasets are very valuable. The motivation for aperiodic TSF and for per-timestep label forecasting is clear. The proposed multimodal approach is principled, and the consistent accuracy gains over baselines support its effectiveness.

**Weaknesses:**

Some concerns:

1. The authors mentioned the Exchange Rate dataset as an exception to periodicity among standard benchmarks, but do not report any experiments on it. All reported experiments are on their two datasets (RP and SimP) for the main per-timestep classification task.

2. To make the proposed EMP method more reliable, the authors can consider evaluating EMP/MP on at least one public aperiodic dataset (e.g., Exchange Rate) to complement the new benchmarks.

3. In the RP/SimP comparisons of EMP and other baselines, please clarify whether any baselines are multimodal. If not, can the authors add a simple multimodal baseline for fair comparison?

4. The authors mentioned that MP outperformed EMP on the RP dataset when P = 18 and attributed this to the possible vision overfitting. Can you substantiate this with any targeted checks?

**Questions:**

See weaknesses.

---

> ### Author Response · Authors · 2025-11-21
>
> * The exchange rate dataset [1] collects daily exchange rates of eight countries from 1990 to 2016. We cite this dataset as an example to illustrate benefits and limitations of existing benchmarks, but it is not suitable for direct evaluation of EMP.  First of all, this time series forecasting dataset only contains numerical values for each day and is clearly collected for regression tasks. Our model is proposed for forecasting classification tasks with multiple data modalities and exogenous variables. Secondly, exchange rate is a dataset in finance dominated by human financial activities, making it very difficult for any predictive model, including ours and prior work. In particular, exchange rate is highly related to science, technology, international polices, agriculture of a country, which is not included in this dataset. One of our main points is that existing datasets lack exogenous variables, which makes long-term prediction less accurate. That is the reason why we spent a lot of effort on collecting our own dataset and models that value other exogenous variables. Simply running our model on a dataset without reasonable exogenous variables makes our model degenerate to previous works and is also against our proposal. Lastly, there are reasons/patterns hidden behind aperiodic data and we want to reproduce this failure multiple times for the critical i.i.d. episode sampling for experiments. However, collecting i.i.d. episodes for the exchange rate dataset is not possible.
> * We are not simply looking for aperiodic datasets. As we illustrated in Appendix C, all the other datasets in the TSF area either maintain their periodicity or show a decreasing autocorrelation all the way to 0. Forecasting a failure needs a model to predict an event at a future time step that breaks the pattern learned from history sequences. This requires a dataset containing periodic and aperiodic data, especially the transition from periodic data to aperiodic data. A qualified dataset should exhibit autocorrelation bouncing up and down and finally converging to 0 indicating little to no correlation and probably unpredictable. However, the exchange rate is such a dataset whose autocorrelation never bounces up.
> * All the other baselines are unimodal. To the best of our knowledge, we proposed the first multimodal model in this area. Considering the limitations of existing time series dataset (unimodal, few or no exogenous variables), it does not surprise us. Other related previous in the CV area are either uni-modal, forecasting the joint angles, 3D positions, or require more modalities like scandots but output walking policies for control (EgoEgo, Marepo, [2]). Proposing a multimodal model in this area and revealing the significance of exogenous variables and multimodalities in the ablation study is part of our contribution in this work.
> * To substantiate this hypothesis, we need to collect data at different locations in person. Maybe we have to increase the number of locations in RP to the same level of SimP, which is thousands of different locations. It implies an extremely labor-heavy data collection even for a professional group. Since we are a small team, we must leave this as future work.
>
>
> [1] Lai, Guokun, et al. "Modeling long-and short-term temporal patterns with deep neural networks." The 41st international ACM SIGIR conference on research & development in information retrieval. 2018.
>
> [2] Agarwal, Ananye, et al. "Legged locomotion in challenging terrains using egocentric vision." Conference on robot learning. PMLR, 2023.

---

### Official Review · Reviewer_wuXv · 2025-11-01

**Soundness:** 2
**Presentation:** 3
**Contribution:** 1
**Rating:** 2
**Confidence:** 4

**Summary:**

The paper introduces a multimodal architecture for per time‑step label forecasting in aperiodic visuo‑motor streams. The task is to predict whether a humanoid robot will have fallen P steps in the future given a short history of egocentric video and proprioception, augmented with known future exogenous inputs (planned joint trajectories). The authors contribute: (i) a fine‑grained TSF classification task distinct from standard regression or whole‑sequence classification; (ii) two benchmark datasets, RP (real Poppy humanoid, 110 episodes at ~16 fps across three indoor locations) and SimP (PyBullet simulation, 2k episodes at 30 fps) with analyses arguing limited periodicity; and (iii) the EMP baseline that merges a DCT‑based motion encoder, a low‑frequency 3D‑Conv vision encoder, and a trajectory encoder that filters top‑k rFFT components.

**Strengths:**

- The problem statement is clearly written and easy to follow, and the overall experimental setup is transparent at a high level

**Weaknesses:**

- The problem being solved is not quite complicated, forecasting whether a robot is going to slip/fall or not N steps into future is a simple classification problem and has been tackled even simply with proprioception (e.g., paper on Predictive Proprioception, IROS 2022).

- It is not understood why a complicated architecture is required for solving this problem.

- Baseline coverage is too narrow; simple methods are missing.The comparison set is restricted to recent deep TSF models (TimesNet, FlowFormer) and an egovision fall detector (EgoFalls). There are no non‑deep baselines (e.g., logistic regression, linear/regularized classifiers, gradient‑boosted trees, or simple temporal CNN/LSTM baselines) trained on engineered features from joint states and known‑future trajectories, exactly the signals that appear most predictive here. Given that MP is already strong, classical models with derivatives of joint angles, short‑window statistics, and trajectory deltas could be competitive and would directly test the claim that deep multimodal fusion is necessary.

- Proprioception‑only literature is under‑engaged. The related‑work section does not engage with prior work that uses proprioception alone for near‑term failure or slip prediction in locomotion (as mentioned above). The manuscript should position its contribution relative to proprioception‑based forecasting and control papers that demonstrate strong predictive signals from joint histories alone, and then justify when and why egovision materially helps this particular forecast.

**Questions:**

See weakness section

---

> ### Author Response · Authors · 2025-11-21
>
> * Forecasting whether a robot is going to fall N steps into the future is tackled in prePARE [1] (could the reviewer please confirm this is the IROS paper they had in mind?). However, we need to point out that all prePARE experiments are done on a quadruped robot with 12 DoF (degrees of freedom), whereas ours is a more challenging bipedal robot with 25 DoF. An often seen review in the robotics area is whether an algorithm applies to different robots (e.g., bipedal robots, wheeled robots, or bionic robots). As far as we know, proposing a general algorithm for as many robots as possible is still challenging in this area. Another question is whether different joints participate in robot walking. Unfortunately, it is true for most cases and therefore leads to a feature space of higher dimension as the DoF increases. prePAREr proposed a shallow baseline for short-term prediction but left its generality without discussion. Besides, all experiments are done on their own imbalanced private datasets without any comparison with other methods (either shallow or deep learning based models), and their code is private as well, making direct comparison difficult - whereas our code and data are both open. Furthermore, bipedal humanoid robots are facing a more difficult stability challenge compared to quadruped robots, which is out of the scope of this paper.
>
> * A simple yet strong motivation for our more complex architecture is to achieve longer-term forecasting. We tested some shallow baselines (see next bullet point) and they are not competitive for longer prediction spans. We also want to incorporate more modalities because whether a slip/fall will happen is determined by the interaction between robots and the surrounding environments. Simply taking joint angles or velocities as inputs exclude the possibility of utilizing information from surrounding environments. Hence, we need a multimodal method not only for this task but also for complicated scenarios.
>
> * We appreciate your suggestion and implemented 2 more shallow baselines (logistic regression and linear classifier) for comparisons. In short, shallow baselines are competitive only in short prediction spans on the RP dataset.  Our model outperforms the shallow baselines in long-term prediction even after dropping the visual clues (i.e. the MP ablation). Our supplementary experiments also show that shallow baselines require dedicated engineered features, which are all dependent on expertise and experience, while our deep models can take care of this by simply taking raw data as inputs. Please refer to the new Appendix section of our manuscript.
>
>   Another interesting finding is that our results verify shallow baselines’ effectiveness on a low ROI (rate of interest, ROI=best prediction span/history sequence length). This metric tells us the inference ability of a given model. The prePARE papers reported a history sequence of 3.6s and reached best performance with prediction length of 0.72s, which indicates an ROI of 0.72/3.6 = 0.2. On our dataset, the shallow baselines we tested also exhibit a decent performance at prediction span 6 indicating an ROI of 6/24=0.25. But, only our EMP model can generalize to longer prediction spans. Please refer to section E in the Appendix for comparisons. We will add more content later.
>
> * We will cite [1] and other recent works in the related work section.
>
> We want to emphasize that our work aims to point out some common disadvantages in the TSF area rather than propose a model for slip/fall prediction that can generalize to all robots and all scenarios. Rather, EMP is intended as a first baseline for our new benchmark datasets to test aperiodic, multimodal time-series forecasting. Hence, we would like to see researchers put more attention on such datasets. In short, outperforming SOTA methods in robotics is not our primary objective, and moreover, predictive tasks on slip/fall for different robots - especially high-DoF humanoids - remains a challenging open problem.
>
> [1] Dey, Sharmita, et al. "PrePARE: Predictive proprioception for agile failure event detection in robotic exploration of extreme terrains." 2022 IEEE/RSJ International Conference on Intelligent Robots and Systems (IROS). IEEE, 2022.

---

> > ### Comment · Reviewer_wuXv · 2025-11-26
> >
> > I thank the authors for their rebuttal. I will keep my original rating since I am yet not able to appreciate the contribution of this work mainly for the following reasons.
> >
> > 1. The work does not present anything substantially new. The problem is not new, the concept of multimodal fusion is not new, and the methodology is essentially a rearrangement of established components. There is no new angle, insight, or perspective. As it stands, it is simply another piece of work in the robotics space without a clear insight or takeaway.
> >
> > 2. I completely support application papers, please do not get me wrong. But even application papers can offer a novelty (in terms of method/perspective/concept/formulation/message) specific to the application; either a design choice, a domain insight, or some adaptation that advances the state of practice. Here, it is just another method with a slightly modified architecture. There is nothing that makes it stand out.
> >
> > 3. Even if the paper were positioned as a benchmarking effort, the baselines are not sufficient. Logistic regression and a linear classifier are far too simplistic, and the rebuttal does not justify why more appropriate deep learning baselines were not included. In short, I do not find any insights in this paper and feel this is just another work w/o a clear contribution/message.

---

> > > ### Author Response · Authors · 2025-11-27
> > >
> > > Please note that we added ensembles of deep baselines in the new Appendix F.

---

### Author Response · Authors · 2025-11-27
**A global response for updates in the rebuttal phase**

This is a global response to all reviewers and ACs. We hope this can make it easy for you to catch all the updates in the rebuttal phase. (Thank reviewer uCay for the kind suggestion.)

All important content is highlighted in red in the most recent version of the manuscript PDF and listed below with line numbers.
1. Line 88: We add a requested citation for an IROS paper in the robotics area.
2. Line 134: We add “where $P\in\mathbb{N}$” at the end of the sentence to clarify that $P$ is a single value (we use different values of $P$ in different experimental configurations, but in each individual experiment $P$ is a single number).
3. Line 240: We add a figure (Figure 8) in Appendix C with some representative frames from the input videos, and reference it in (sub-)subsection 3.2.2.
4. Line 272: We add “Note that motion plan can be periodic, whereas the actual observed joint positions, visual input, and importantly the fall/stand labels, are not periodic” to further emphasize the very limited periodicity in our dataset.
5. Line 365: We add text in subsection 4.3 explaining that the comparison is not completely fair because no existing baselines were perfectly suited to our dataset, but these were the closest we could find.  We also clarify in the text exactly which modalities were provided to which baselines.
6. Line 484-485: We agree the RP size is a significant limitation, but given the time-intensive nature of physical robot data collection we are not able to address this before the rebuttal window closes. We add a sentence to the conclusion emphasizing the importance of more data collection in future work.
7. Line 484-485: In principle, EMP’s motion and trajectory encoders can work with any numeric time series arranged in NxT arrays, so there is potential it might work on other non-robotic data that combines video and lower-dimensional numerical measurement vectors. Of course, that hypothesis is not addressed by our current experiments.  We add this as another direction for future work in our conclusion.
8. Line 692: We add more details in Appendix C that emphasize the substantial differences in motion diversity between RP and SimP and a scatter plot (Figure 7) from birds-eye view of final robot poses in SimP to highlight this distinction.
9. Line 701: We take special care to ensure the model was not simply a persistence model and add details in Appendix C along with a plot (Figure 9) showing representative head $z$ coordinate time-series and where the prediction span time range sits along those time series.
10. Line 805 and 858: Considering the suggestion from wuXv, we implemented 2 more shallow baselines (logistic regression and linear classifier) for comparisons (Table 7) in Appendix E. And also report an interesting finding in this section.
11. Line 884: A potentially fairer comparison is added in Appendix F. We use three ensembles of the three baselines (TimesNet, FlowFormer and Egofalls), which effectively becomes a multimodal predictor. The performance is reported in Table 8.

Short summary:
* Point 1, 10 address several issues raised by reviewer wuXv.
* Point 5 addresses several issues raised by reviewer bxW8.
* Point 2, 3 address several issues raised by reviewer p6gi.
* Point 4, 5, 6, 7, 8, 9, 11 address several issues raised by reviewer uCay.
* Please see our official comments to each reviewer for the remaining issues.

Here are minor modifications that do not introduce new content and rewording for saving space. They are not highlighted in red.
1. Line 108: The title of subsection 2.3 is changed to “Multimodal TSF”.
2. Line 240: "...which imitates how human beings walk in the real world" -> "which imitates human beings walk." We reword this sentence to save space.
3. Line 308: "...are used for training and validation. Models are tested on the hallway data." -> "...are used for training and validation, and hallway for testing." We reword this sentence to save space
4. Line 460: We merge two paragraphs to save space without any rewording. “We test two ablations of the input data provided to…” is the first sentence of the next paragraph before merging.
5. Line 472: The title of section 5 is changed to “Conclusion and Future Work”.

---

### Meta-Review · Area_Chair_pbLU · 2026-01-08

**Summary:**

Reviewer wuXv

(1) Novelty is limited (as all components of the method have been previously observed in prior work); (2) the work is an application paper with few takeaways for other applications or for theory; (3) no nontrivial baselines (only logistic regression and linear classifier).

Reviewer bxW8

(1) A canonical aperiodic dataset is mentioned but not tested on (Exchange Rate); (2) at least one existing public aperiodic dataset (like Exchange Rate) should have been tested on; (3) simple multimodal baseline is missing; (4) a baseline outperforming the proposed method is deflected due to unsubstantiated claims of overfitting.

Reviewer p6gi

(1) Novelty of the work as a time series classification or anomaly detection method is limited; (2) a multimodal baseline like Time-VLM is missing; (3) citations to the related areas of multimodal action recognition and motion prediction are missing; (4) methodological details, including about the vision sequences, are missing from the text.

Reviewer uCay

(1) There are inconsistencies between the goal of the work (aperiodicity) and the design of a component (the trajectory encoder); (2) baselines are trivial because they receive less information than the proposed method; (3) components of the proposed method don't seem to be uniformly helpful; (4) clarity about what is actually forecast is insufficient; (5) dataset diversity is limited.

**Reviewer Concerns:**

Addressed

Reviewer p6gi (4)

Methodological details, including about the vision sequences, were addressed.

Outstanding

Reviewer wuXv (1), Reviewer p6gi (1)

Concerns about the novelty of the method remained unaddressed.

Reviewer wuXv (2)

Concerns about relevance for the broader ICLR audience remained unaddressed.

Reviewer wuXv (3), Reviewer bxW8 (3), Reviewer p6gi (2), Reviewer uCay (2)

Concerns about the sufficiency of the empirical baselines if the main contribution is a new method were not addressed.

**Reviewer Scores:**

I believe Reviewer wuXv and Reviewer p6gi would have maintained their low scores, as the concerns about the novelty of the method, its relevance for the broader ICLR audience, and the sufficiency of the empirical baselines if the main contribution is a new method, were not addressed.

Reviewer bxW8 was already mildly positive, but did not reflect on any of the significant concerns from Reviewers wuXv and p6gi.

Reviewer uCay was willing to update their score, and I believe would have updated their score to 6.

This paper is thus borderline, but I assess that the significance of the concerns from Reviewers wuXv and p6gi outweigh the positivity of the remaining reviewers.

---

### Decision · Program_Chairs · 2026-01-26

Reject